# Variations in the constituent year effect in Junior World Championships in alpine skiing: A window into relative development effects?

Øyvind Bjerke[1]*, Håvard Lorås[1,2], Arve Vorland Pedersen[3]

**1** Department of Teacher Education, Norwegian University of Science and Technology, Trondheim, Norway, **2** Faculty of Education and Arts, Nord University, Levanger, Norway, **3** Department of Neuromedicine and Movement Science, Norwegian University of Science and Technology, Trondheim, Norway

* oyvind.bjerke@ntnu.no

**Data Availability Statement:** All data is publicly available at the Fédération Internationale de Ski (FIS) website: www.fis-ski.com, and can be accessed by anyone through any regular web

## Abstract

While research on the effects of 'birth month' is usually referred to as relative age effects, the study of the effects of 'birth year' is described as the constituent year effect (CYE). In the present study we examined the impact of the CYE on participation in the Junior World Championship in alpine skiing. Based on previous research, we expected to find increasing numbers of participants the older the age-group, and that the CYE would be stronger in the speed events compared to the technical ones. The sample in the present study consisted of 1188 male skiers and 859 female skiers within the age range of 17 to 21 years at the time of competition. The results show that the number of male participants increased with increasing age, which can be described as a CYE. For female skiers, a CYE was found, but it dissipated two years earlier than for male skiers. The CYE varied with event and was more pronounced the higher the speed of the event. The findings thus suggest that a constituent year effect exists among skiers participating in the FIS Junior World Ski Championship in the alpine skiing championships, and that the effect varies with gender and event, rather unrelated to age. Thus, it seems that the effect may not be a relative age effect, but instead a relative development effect.

## Introduction

In recent years, a vast amount of literature has been produced on the relative age effect (RAE) in sports or terms related to this topic [shown in a number of studies, e.g. 1]. The term RAE denotes the "overall difference in age between individuals within each age group" [2]. Grouping children chronologically by age induces a potentially large difference in age, which has shown to have a great impact on selection biases in sports competitions as well as affecting school results [3, 4]. The RAE was first described by Barnsley et al. [5] in ice hockey, whilst the effect attracted public interest after it was discussed in Malcolm Gladwell's *Outliers* [6]. The RAE has been detected, and most often reported, in team sports like soccer [2, 7], ice hockey [8], and handball [9, 10]. In addition, the effect has been found in individual sports like swimming and tennis [11], and in alpine skiing [12–14]. The RAEs are more pronounced in sports

browser. Thus, the authors did not have any special access privileges. All data is collected and available at this website: www.fis.com.

**Funding:** The authors received no specific funding for this work.

**Competing interests:** The authors have declared that no competing interests exist.

that are deemed culturally important due to stronger selection pressure, like alpine skiing in Austria [7, 14, 15]. Sometimes variations in the RAE have been described, such as an inverse relative age effect [9], indicating that there is an advantage in being born late in a cohort. Some researchers even claim that relatively younger rugby players selected for a talent program during adolescence have a greater possibility of future career success [16].

The main mechanism underlying the RAE is usually suggested to be the physical advantage of being born earlier compared to the remaining members of the same cohort, commonly referred to as the maturation-selection hypothesis [7]. Furthermore, being born up to one year earlier than one's peers allows time for extra practice, which can be a large percentage of the child's total practice time in that particular sport. In turn, the initial developmental advantages will affect sociological and psychological processes and advantages, which are explained and related to the Pygmalion effect and self-fulfilling prophecy (see Musch & Grondin, 2001 for a review). Another mechanism that could also explain the increasing advantage is the Matthew effect, which means that for individuals who gain an advantage not possessed by many of their peers, these advantages will accumulate over time (see Hancock et al. [17]).

While these secondary effects (notably, the Pygmalion effect and Matthew effect) have been studied in detail, also with testing of the theoretical tenets of the concepts, the RAE itself seems so far to have evaded any real explanation and could be described as rather "atheoretical" [18]. Studies on the RAE have mainly reported the occurrence of the effect within a steadily increasing number of domains, while its variations have received less attention. In order to better understand the mechanisms that interact in producing the RAE, Wattie et al. [18] proposed a theoretical model grounded upon Newell's [19] framework of interacting constraints (individual, environmental and task). Newell based his work on dynamical systems theory [20], a principle which Thelen also attempted to incorporate in her theory of development [21, 22].

The usual understanding of the effects of relative age includes age differences within a given annual year, also known as the within one year effect [23]. Less known is the constituent year effect (CYE), which concerns the effects of relative age when several within-one-year cohorts participate in multiyear age bands, i.e. 16 to 19 years of age. When multiyear age bands are kept constant across development, the effect is termed the constant year effect [23]. Such effects, were originally studied in Masters sports, where the cohort grouping often has a five-year band [24]. While youth systems often show an overrepresentation in the oldest age cohort, the opposite effect is typically evident within the Masters sports. If the age band works over a larger age span, the impact of the constituent year effect is stronger [25]. The relative age effect is undoubtedly present among the youngest athletes, and at a particular point, the effect will diminish (and even turn), at least typically at the Masters level. In fact, studies performed on Masters athletes indicate that performance declines with age after age-related peak performance due to reduced physical capabilities [24]. In a systematic review of age of peak performance, Allen et al. [26] found that in explosive power/sprint sports, the peak performance is around 27 years, while the age for peak performance increases in endurance sports.

In alpine skiing, the RAE has been reported in several studies [13, 14, 27]. The effect has been well documented among youth alpine skiers in Youth Olympics [28] and in skiers at the World Cup level [13]. However, among the best World Cup alpine skiers there are variations in relative age effects according to discipline and sex [29]. At the absolute top level there seems to be an inversion of the relative age effect, as the relatively younger skiers perform at a higher level, collecting on average more World Cup points [30]. As the effects of relative age are well known and thus less interesting per se, merely reporting the effects in various sports or other domains may not advance knowledge of the RAE very much or bring us closer to an understanding of the mechanisms underlying the effect. Even though the effect is well known, nothing indicates that the effects have disappeared in alpine skiing in the last decade [31].

If the RAE were mostly about maturation (initial advantage) compared to, e.g. extra practice, it would be more prominent in the speed discipline, where the effect of body size is largest, as indicated by Bjerke et al. [29]. If, on the other hand, the RAE were more about practice, or the Pygmalion or Matthew effects, we would see a more even distribution of birth years across disciplines, as these effects would be more similarly distributed among those who were initially selected and would enhance differences to a lesser degree.

Both the RAE and CYE lend themselves well to scientific studies, as data are available in abundance on the internet. However, as differences and variations may be subtle, very large datasets would be required in order to study the RAE properly. The CYE would work in similar ways to the RAE, but it would be even more pronounced, as it works on a larger time span. Individuals are competing, and are thus compared across intervals of two years, or even five years, as in the Masters case.

Steingröver et al. [25] found a constituent year effect among elite German youth basketball players, where the age band ranges from 13 to 16 years (JBBL), but such effects were not evident among players of 17 to 19 years of age (NBBL players). In their study, only 5% of JBBL players belonged to the youngest age cohort, which was explained by the maturational differences across the age band, as the players enter the talent development system with a delay. Similar results were found in soccer [32], where data from soccer players under the age of 17 were analysed. In another study, Cobley et al. [33] found the constituent year effect to be evident among males between 12 to 15 years of age. In this study, at the age of 15 to 16 years, the effects had dissipated, and at the age of 17 to 18 years, the effects actually invert, so that the relatively younger athletes are more represented within the actual age band. These studies show a variety in how the CYE appears in sports. Examination of individual sports is claimed to be valuable to uncover the mechanisms of the RAE since variables that may confound the effect are easier to identify [34].

The aim of the present study was to investigate whether the constituent year effect would vary in ways similar to the relative age effect. The present data were collected from the Youth World Championship in all events in alpine skiing, and included both sexes. The athletes were from 17 to 21 years old, and the effects were studied across all alpine skiing events. Based on previous research, the hypothesis was that the CYE would be evident by more participating skiers being born early in the cohort, and more pronounced in males than in females. Furthermore, it was hypothesized that the CYE has a greater impact in the speed disciplines compared to the technical disciplines.

## Methods

### Participants

The male and female alpine skiers participating in the Fédération Internationale de Ski (FIS) Junior World Ski Championship in alpine skiing in Sochi 2016, Åre 2017, and Davos 2018 were selected for the present study. This comprised an overall sample of 1188 male skiers and 859 female skiers within an age range of 17 to 21 years at the time of competition.

### Study variables

Data were obtained from the Fédération Internationale de Ski (FIS) website (www.fis-ski.com [35]), and included the skiers' year of birth, sex and events (slalom, giant slalom, super-G and downhill). The year of birth was applied to compute the age of each skier at the time of competition.

## Statistical analysis

In order to examine the constituent year effects, the distribution of the participants across age, sex and events was examined by Chi-square tests ($\chi2$) against an even distribution, with Cramer's $V$ ($\phi$) as a measure of effect size interpreted according to Cohen [36] as small <0.05, medium ≥0.06 to <0.24, and large ≥0.25. Predictive Analytics Software (PASW, IBM, NY, US; previously SPSS) Version 25.0.0.1 was used for all statistical procedures with $p < 0.05$ as statistical significance criterion.

## Results

### Male junior alpine skiers

As is clearly visible in Fig 1, the younger male junior alpine skiers were substantially under-represented in the 2016–2018 World Ski Championships ($\chi2$ = 408.89, $df$ = 4, $p < .0001$, $\phi$ = 0.59). This pattern of results was also found across the four events ($\chi2 \geq 87.30$ $df$ = 4, $p < .0001$, $\phi \geq 0.49$). Furthermore, the proportion of 17-year-old male skiers was systematically under-represented compared to the older skiers ($\chi2 \geq 4.77$, $df$ = 1, $p \leq 0.029$, $\phi \geq 0.26$), the proportion of 18-year-old male skiers was systematically under-represented compared to the older skiers ($\chi2 \geq 5.24$, $df$ = 1, $p \leq 0.022$, $\phi \geq 0.22$), and the proportion of 19-year old male skiers was systematically under-represented compared to the 20-21-year-old skiers ($\chi2 \geq 12.07$ $df$ = 1, $p \leq 0.0005$, $\phi \geq 0.26$). However, no significant differences were found in the proportion of 20-year-old compared to 21-year-old skiers ($\chi2 \leq 0.29$ $df$ = 1, $p \geq 0.59$, $\phi \leq 0.04$). These latter patterns of results were similar across the four events.

### Female junior alpine skiers

As can be seen in Fig 2, examination of the female junior alpine skiers' age at competition indicated a certain degree of over-representation of older skiers. Indeed, the Chi-square analysis indicated a significant overall difference in frequencies across the ages in female skiers ($\chi2$ = 68.28 $df$ = 4, $p < .0001$, $\phi$ = 0.28). Conducting the analysis event-by-event, the distribution across age at competition in slalom was significantly different from an even distribution ($\chi2$ = 10.36 $df$ = 4, $p$ = 0.035, $\phi$ = 0.20). A similar pattern of results was observed in giant slalom ($\chi2$

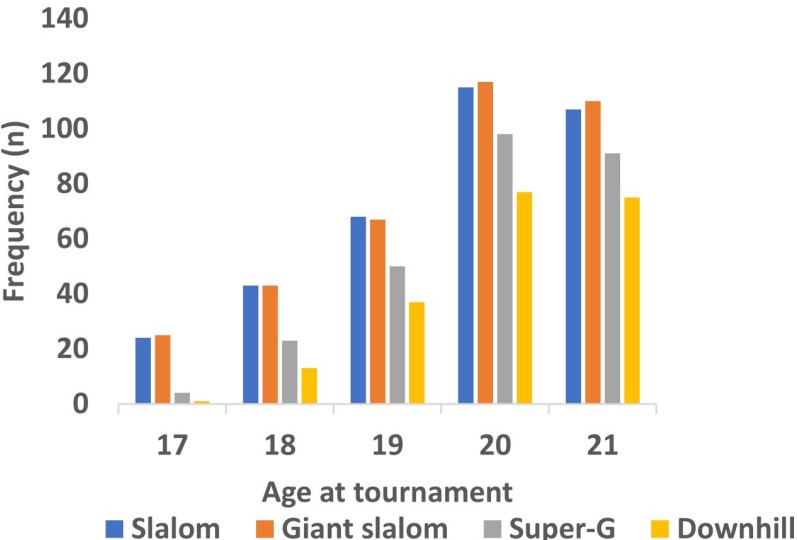

**Fig 1. Frequency of participants across age cohorts and events among male alpine skiers participating in the 2016–2018 Junior World Championships ($n$ = 1188).**

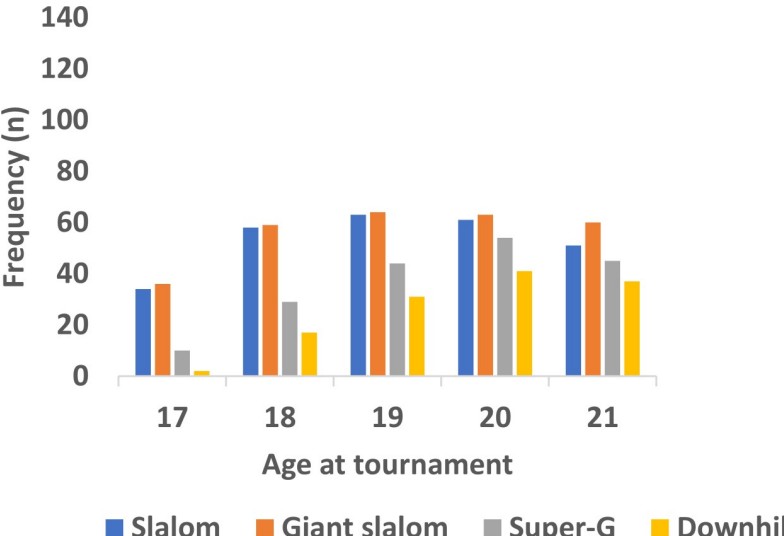

**Fig 2. Frequency of participants across age cohorts and events among female alpine skiers participating in the 2016–2018 Junior World Championships ($n$ = 859).**

= 9.53 $df$ = 4, $p$ = 0.0492, $\phi$ = 0.18), super-G ($\chi2$ = 32.78 $df$ = 4, $p$ < .0001, $\phi$ = 0.42), and downhill ($\chi2$ = 40.13 $df$ = 4, $p$ < .0001, $\phi$ = 0.56). Across events, the proportion of 17-year-old female skiers was under-represented compared to the older skiers ($\chi2 \geq 6.26$ $df$ = 1, $p$ < 0.01, $\phi \geq$ 0.26). Conducting further analysis across the 18-21-year-old skiers, no significant differences in distributions were found in either slalom or giant slalom ($\chi2 \leq 1.42$ $df$ = 3, $p \geq 0.70$, $\phi \leq$ 0.08). In super-G and downhill, the distributions across 18-21-year-old female skiers were significantly different from an even distribution ($\chi2 \geq 32.13$ $df$ = 3, $p$ < .0001, $\phi \geq 0.48$).

## Discussion

The results of the present study show a clear and strong CYE among male skiers, in that the number of participants increased with increasing age up to 20 years in the FIS Junior World Ski Championship in alpine skiing. Thereafter, the CYE was not evident, as no difference was found between 20 and 21-year-olds. For the female skiers, a difference (CYE) was found between 17- and 18-year-olds, in that more 18-year-olds participated. After the age of 18, there were no differences (thus, no CYE) between any age groups. We should keep in mind that the present results include only *participation* in the Junior World Championship, and not their actual results. However, all the skiers were selected by their national federations to represent their countries in these competitions. Thus, they were judged to be worthy of such participation in (sometimes fierce) competition with other athletes, who may have been up to four years older. It would be reasonable to assume that their selection would reflect their chances of a reasonable result in competition with the other, similarly selected, participants.

The present findings were not surprising, and they fall in line with predictions based on previous findings on the CYE and the RAE [24, 37]. This was, however, not the scope of the current paper, other than laying the ground for our second predictions and acting as a check that our data are valid for further exploration.

As was suggested in the introduction of the present paper, the CYE seems to provide a sort of magnifying lens on relative age effects making any variation more clearly visible. The CYE is, in principle, the same effect as the RAE; in fact, the CYE *is a RAE*, only stronger, since it

defines RAE more broadly over a longer time span [1]. This makes it, as mentioned above, ideal for studying smaller variations within the bigger picture of the effect.

Bjerke [29] showed an effect of discipline (technique vs. speed) within the general RAE in male alpine skiers at the highest level that on by closer inspection turned out to be an opposite or inverse RAE [30]. The RAE was present in the speed disciplines (super-G and downhill racing), but not in the technical disciplines (slalom and giant slalom). In that study, no effect whatsoever was found among female alpine skiers.

In the present dataset, the results show a neat and orderly effect of discipline within the overall CYE, in that there were almost linearly decreasing numbers of participants with increasing speed in the event (in the order: slalom, giant slalom, super-G, downhill), irrespective of age. The age trend was similar across events, but grew stronger with the increasing speed of the event; hence, it can be argued to be an effect of speed. This trend was evident in both males and females (Figs 1 and 2). In fact, for males, there were almost equally as many 17-year-old participants in the slalom competition as there were 21-year-olds in the downhill competition. When remembering that the overall trend was the expected effect of (relative) age, both in males and females, our data support the general RAE/CYE [shown in a number of studies; e.g. 1, 9, 14], as well as the variations of the effect with discipline, as shown by Bjerke [29]. However, in the present dataset, these variations can be studied in more detail, and the data show that in addition to the differences between technique and speed events, there is an almost linear relationship between the relative age of participants and the speed of the event.

Had the RAE (CYE) resulted merely from maturation, and not from extra practice or additional effects (like the Pygmalion effect or the Matthew effect) affecting the initial RAE, one would expect the effects to disappear at a more similar rate across events after the maturation difference had been equalled out. On the other hand, if the effect was due to an "extra practice" effect, the difference would persist longer, as it is hard to compensate for up to one year more of practice. Also, in that case, the effect would be more similar across events.

The most common explanation of the RAE is the maturation hypothesis, favouring the early-born who are, on average, stronger and larger than their peers [7, 38]. Research on alpine skiing has demonstrated that relatively older athletes have an increased likelihood of being selected for teams if they are taller and heavier [27]. This takes place despite the fact that there are no differences in physical motor skills among ski racers at an adolescent age (14–15 years), indicating that there are other mechanisms responsible for the difference in skiing performance as the skiers get older [39]. It has been shown that the RAE dissipates or disappears in the longer term, and even an inverse effect has been reported among the oldest cohorts [16, 30, 33]. The disappearance of the effect is usually explained by the equalization of physical factors like body size, weight and height [33], and by the "underdog effect" [40].

In a typical winter sport like alpine skiing, the co-occurrence of the beginning of the calendar year and the winter (hence the alpine skiing) season would enhance the advantage of being early-born relative to many other sports. For the CYE, however, when the (relative) age effect works over a longer time span (five years), there would be no such overall advantage of the season, and thus the "winter-effect" would be eliminated. Another possible explanation was put forward by Moreira [37], who argued that relatively younger athletes may have a poorer motivational orientation since they are competing against more established skiers and hence have a lower expectation of success in relation to their peers.

The present results show that there is a point of culmination, and this varies across sex and event. Previous research on the RAE explains such differences across sexes, with male sports being more competitive due to the higher number of participants [41, 42]. The CYE among female skiers in the present dataset disappeared between the ages of 18 and 19 years, and no difference in the numbers of participants was present from 18 to 21 years of age. For males, the

CYE lasted two years longer, and differences in participation were demonstrated between all age groups up to 20 years. A two-year difference would coincide reasonably well with the differences in the onset of puberty between the two sexes [43] and would further suggest that the advantages of puberty-related variables would be greater in boys. During puberty, the height and lean body mass increase, resulting in further anthropometric and physical advantages [33]. Speaking against the puberty explanation is the fact that the effect of the event was somewhat independent of age and was relatively similar across sexes.

The Matthew effect [17] seems to be somewhat institutionalized in alpine skiing, as there is a ranking system based on FIS points, which benefits those skiers who have been in the FIS points system over time. The best-ranked skiers are given the most favourable starting positions, and hence the best-prepared course, whereas the youngest skiers are disfavoured in the system. The advantage of the ranking system will accumulate over time, corresponding to a Matthew effect in starting position.

The current study illuminates some of the complexity and directs some explanations of the RAE and CYE. Additionally, it shows that further insight into the variations in RAE/CYE is needed. That the existing CYE follows the general trend of the RAE, but at the same time is stronger, would make it relevant to also speculate whether also the same variations with the event would be found within the RAE, as would also be suggested by the mentioned findings of Bjerke et al. [29]. In fact, based on the results in this study, there is reason to suppose that the same mechanism could be present in the RAE. In order to reduce the relative age effects several possible solutions are suggested, e.g. rotating cut-off date and expanding the age bands, using competition groups based on height and weight or delaying the process of talent identification [7]. All these are based on the assumption that age itself, or the relative birth date, is in fact the cause of the discussed problems with the RAE. The present results indicate that such solutions may not be particularly appropriate, as age per se is not the relevant constraint or underlying mechanism.

As was proposed by Wattie et al. [18], the study of the RAE has appeared somewhat "atheoretical", lacking explanations of the phenomenon outside stating that those born early in the year are on average bigger and stronger. The present data has several features that make them better suited for testing the theoretical tenets of the RAE and possibly explain the variations by other factors than age per se. Firstly, as mentioned, the present CYE works over five years instead of one year as is usual for the RAE. Secondly, variations of the effect can be studied across four events instead of between two disciplines (speed and technique) as in Bjerke et al. [29]. Finally, by including both sexes, it is possible to study the effect of physical development (notably puberty) since the development is similar across sexes only skewed by some two years. Studying the latter was made possible by means of the "magnifying effect" of the CYE relative to the RAE, as the RAE has usually been small in female skiers; hence, it has been less frequently studied compared with males.

The present data, thus, suggest that relative age may not be the real constraint on performance, but a mere proxy. There was an effect of age, but this was not equal for the different events; it was even almost linearly related to the speed of the event. Furthermore, age effects varied with sex, however systematically skewed with about two years suggesting that sex was not the relevant variable but instead the relative difference in development (puberty) was. Thus, from a constraints-based perspective [19], it is possible to argue, with Wattie et al. [18], that relative age may not in fact be the relevant constraint, but that relative development is. From this, one could argue that the RAE should rather be called the relative development effect (RDE).

## Limitations

The skiers participating in the various junior World Championships, included in the analysed sample, are selected by their respective national ski-federations. The nature of the selection process within each federation is not known, and there might be many types of considerations involved in such a process. Given that each nation can register a limited number of skiers for the championship, federations might nominate skiers evaluated to be candidates for winning medals or simply for gaining experience in tournaments at the highest international level.

Also, competition results from the various championships have not been analysed in relation to the constituent year effect. Further studies including this analysis might indicate whether the oldest skiers are over-represented amongst the highest-ranked skiers, which could provide another line of evidence for relative age effects in junior elite alpine skiing.

## Author Contributions

**Conceptualization:** Øyvind Bjerke, Håvard Lorås, Arve Vorland Pedersen.

**Data curation:** Øyvind Bjerke, Håvard Lorås, Arve Vorland Pedersen.

**Formal analysis:** Øyvind Bjerke, Håvard Lorås, Arve Vorland Pedersen.

**Investigation:** Øyvind Bjerke, Håvard Lorås, Arve Vorland Pedersen.

**Methodology:** Øyvind Bjerke, Håvard Lorås, Arve Vorland Pedersen.

**Project administration:** Øyvind Bjerke, Håvard Lorås, Arve Vorland Pedersen.

**Resources:** Øyvind Bjerke, Håvard Lorås, Arve Vorland Pedersen.

**Software:** Øyvind Bjerke, Håvard Lorås, Arve Vorland Pedersen.

**Supervision:** Øyvind Bjerke, Håvard Lorås, Arve Vorland Pedersen.

**Validation:** Øyvind Bjerke, Håvard Lorås, Arve Vorland Pedersen.

**Visualization:** Øyvind Bjerke, Håvard Lorås, Arve Vorland Pedersen.

**Writing – original draft:** Øyvind Bjerke, Håvard Lorås, Arve Vorland Pedersen.

**Writing – review & editing:** Øyvind Bjerke, Håvard Lorås, Arve Vorland Pedersen.

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
