## [Decision Letter · Decision Letter 0]

24 Dec 2019

PONE-D-19-22259

Variations in the Constituent Year Effect in Junior World Championships in Alpine Skiing

PLOS ONE

Dear Mr. Bjerke,

Thank you for submitting your manuscript to PLOS ONE. After careful consideration, we feel that it has merit but does not fully meet PLOS ONE’s publication criteria as it currently stands. Therefore, we invite you to submit a revised version of the manuscript that addresses the points raised during the review process.

Please address the reviewers comments in a point by point manner.

We would appreciate receiving your revised manuscript by Feb 01 2020 11:59PM. To enhance the reproducibility of your results, we recommend that if applicable you deposit your laboratory protocols in protocols.io, where a protocol can be assigned its own identifier (DOI) such that it can be cited independently in the future. For instructions see: http://journals.plos.org/plosone/s/submission-guidelines#loc-laboratory-protocols

We look forward to receiving your revised manuscript.

Kind regards,

Caroline Sunderland

Academic Editor

PLOS ONE

Journal Requirements:

Please ensure that your manuscript meets PLOS ONE's style requirements, including those for file naming. The PLOS ONE style templates can be found at http://www.plosone.org/attachments/PLOSOne_formatting_sample_main_body.pdf and http://www.plosone.org/attachments/PLOSOne_formatting_sample_title_authors_affiliations.pdf

Reviewers' comments:

Reviewer's Responses to Questions

**Comments to the Author**

1. Is the manuscript technically sound, and do the data support the conclusions?

Reviewer #1: Yes

Reviewer #2: Yes

2. Has the statistical analysis been performed appropriately and rigorously? 

Reviewer #1: Yes

Reviewer #2: Yes

3. Have the authors made all data underlying the findings in their manuscript fully available?

Reviewer #1: Yes

Reviewer #2: Yes

4. Is the manuscript presented in an intelligible fashion and written in standard English?

Reviewer #1: Yes

Reviewer #2: Yes

5. Review Comments to the Author

Reviewer #1: GENERAL COMMENTS

This study examined the impact of the Constituent Year Effect (CYE) on participation in the junior World Championship in alpine skiing, considering sex and discipline as mediating variables. The results showed that the number of male participants increased with increasing age, fact evidencing CYE among male skiers. For the female skiers, a difference (CYE) occurred between 17- and 18-year-olds. Furthermore, the CYE varied with disciplines and was more pronounced in speed disciplines. These findings provide new insights about the influence of the Constituent Year Effect (CYE) and Relative Age Effect (RAE) in alpine skiing context.

In my opinion, this study is interesting and meets the minimum requirements for publication in the Plos One. However, in the current version, I have some minor concerns. In the following I have listed my suggestions.

Introduction

Generally, the authors have written a good introduction that sets the 'scene' for the authors. I have only a few suggestions regarding the theoretical background.

1) The authors could explore recent research on relative age effect.

2) In addition, Wattie et al. (2014) presents an interesting theoretical model that can help support the study hypotheses. (DOI 10.1007/s40279-014-0248-9)

Methods

Statistical analyses

Please identify what your classifications were for interpreting effect size.

Results

I really think the pictures are of poor quality. Please use specific software to build graphics and improve overall quality.

Discussion

The discussion addresses the main points of the data and interpretation. I simply question the practical implications of these findings. I think that the authors could take a small approach about this.

References

Please, make sure you follow the formatting guidelines for references. Some misconceptions were seen in the following references: 4, 18, 21, 23, 28, 35.

Reviewer #2: In order to meet the minimum character count requirement for this section, I am including random characters below. Please see my attached comments in the submitted document. Thanks,

xxxxxxxxxxxxxxxx

6. PLOS authors have the option to publish the peer review history of their article (what does this mean?). If published, this will include your full peer review and any attached files.

Reviewer #1: No

Reviewer #2: Yes: Ajit Korgaokar

---

## [Author Response · Author response to Decision Letter 0]

2 Feb 2020

We have a point to point response document attached for both reviewers.

---

## [Decision Letter · Decision Letter 1]

24 Mar 2020

Variations in the Constituent Year Effect in Junior World Championships in Alpine Skiing - a window into Relative Devolopment Effects?

PONE-D-19-22259R1

Dear Dr. Bjerke,

We are pleased to inform you that your manuscript has been judged scientifically suitable for publication and will be formally accepted for publication once it complies with all outstanding technical requirements.

With kind regards,

Caroline Sunderland

Academic Editor

PLOS ONE

Reviewers' comments:

Reviewer's Responses to Questions

**Comments to the Author**

1. If the authors have adequately addressed your comments raised in a previous round of review and you feel that this manuscript is now acceptable for publication, you may indicate that here to bypass the “Comments to the Author” section, enter your conflict of interest statement in the “Confidential to Editor” section, and submit your "Accept" recommendation.

Reviewer #1: All comments have been addressed

Reviewer #2: All comments have been addressed

2. Is the manuscript technically sound, and do the data support the conclusions?

Reviewer #1: (No Response)

Reviewer #2: Yes

3. Has the statistical analysis been performed appropriately and rigorously? 

Reviewer #1: (No Response)

Reviewer #2: Yes

4. Have the authors made all data underlying the findings in their manuscript fully available?

Reviewer #1: (No Response)

Reviewer #2: Yes

5. Is the manuscript presented in an intelligible fashion and written in standard English?

Reviewer #1: (No Response)

Reviewer #2: Yes

6. Review Comments to the Author

Reviewer #1: (No Response)

Reviewer #2: (No Response)

7. PLOS authors have the option to publish the peer review history of their article (what does this mean?). If published, this will include your full peer review and any attached files.

Reviewer #1: No

Reviewer #2: Yes: Ajit Korgaokar

---

## [Editor Report · Acceptance letter]

7 Apr 2020

PONE-D-19-22259R1 

Variations in the Constituent Year Effect in Junior World Championships in Alpine Skiing - a window into Relative Development Effects? 

Dear Dr. Bjerke:

I am pleased to inform you that your manuscript has been deemed suitable for publication in PLOS ONE. Congratulations! Your manuscript is now with our production department. 

With kind regards,

on behalf of

Dr. Caroline Sunderland 

Academic Editor

PLOS ONE